# Directly Controlling the Transport Properties of All-Nitride Josephson Junctions by N-Vacancy Defects

**DOI:** 10.3390/nano13030542

**Published:** 2023-01-29

**Authors:** Junling Qiu, Huihui Sun, Yibin Hu, Shuya Wang, Chuanbing Han, Zheng Shan

**Affiliations:** 1State Key Laboratory of Mathematical Engineering and Advanced Computing, Zhengzhou 450001, China; 2Hongzhiwei Technology (Shanghai) Co., Ltd., Shanghai 200120, China

**Keywords:** defect, quantum transport, NEGF, all-nitride Josephson junction

## Abstract

All-nitride Josephson junctions are being actively explored for applications in superconducting quantum chips because of their unique advantages including their antioxidant chemical stability and high crystal quality. However, the theoretical research on their microstructure mechanism that determines transport properties is still absent, especially on the defects. In this paper, we apply the first principles and non-equilibrium Green’s function to calculate the electrical transport characteristics of the yellow preset model. It is first revealed that the N-vacancy defects play a crucial role in determining the conductivity of the NbN-based Josephson junctions, and demonstrate the importance for the uniformity of vacancy distribution. It is found that the uniform number of vacancies can effectively increase the conductance of Josephson junction, but the position distribution of vacancies has little effect on the conductance. The work clarifies the effect of the N-vacancy defects on the conductivity of the NbN-based Josephson junctions, which offers useful guidance for understanding the microscope mechanism of the NbN-based Josephson junction, thus showing a great prospect in the improvement of the yield of superconducting quantum chips in the future.

## 1. Introduction

Superconducting quantum chips have shown the unique application potential in superconducting quantum computers. The motivation in developing superconducting quantum chips comes from the increase of their decoherence time by the improvement in circuit design [1], new material choices [2,3,4,5,6,7,8], and the optimization of fabrication process [9,10]. As an important component of superconducting quantum chips, the choice of materials for Josephson junctions (JJs) affects the performance. In particular, epitaxially grown all-nitride Josephson junctions exhibit great potential to solve material-related problems, such as reducing microscopic two-level fluctuations because of their antioxidant chemical stability and high crystal quality [9]. Among them, niobium nitride (NbN)/aluminum nitride (AlN)/NbN JJ has been widely studied [3,4,11,12]. AlN, as the insulating layer, has the characteristics of a wide band gap and high resisitivity, while NbN, as the electrode material, has a high superconducting gap (~5.2 meV) and critical temperature (Tc = 17K), which suppresses the excitation of quasiparticles [9,13,14].

However, the properties of this all-nitride tunnel junction strongly depend on the crystal characteristics of the NbN and the interface around the tunnel barrier. Therefore, the NbN layers with a single crystal structure and a clean interface with the tunnel barrier are the necessary conditions for the development of high-quality all-nitride JJs [15]. To date, good experimental results have been achieved by optimizing the substrates and process conditions [9,12,14,15,16,17,18,19,20]. For example, through increasing the buffer layers between the junction and the substrate to reduce the dielectric loss, the energy relaxation time T1 of the all-nitride junctions reaches 16 microseconds (ms), and the phase relaxation time T2 reaches 22 ms [13,21]. However, the results still fall short of expectations.

Recently, great efforts have been focused on exploring the simulation for the microstructure of all-nitride Josephson junctions owing to the close relationship between the physical properties and the electronic structure of the junction [22], and expected to help improve their performance. For example, John et al. [23] used density functional theory to study the microstructure and electrical properties of all-nitrogen interlayers, and the density of states revealed the causes of metallicity and rigidity in the interlayer. Zhang et al. [24] studied the growth mechanism of AlN by molecular dynamics. The effects of different N:Al fluxes ratios, temperatures, and stresses on the crystallinity of AlN were analyzed, and the environmental conditions of the best crystallization quality and the location of defects were obtained. However, the electrical characteristics of the microstructure in all-nitride JJs have not been described qualitatively/semi-quantitatively by computer simulation, especially with defects. Therefore, to make better use of the inherent advantages about all-nitride JJs, it is worth conducting further research.

In this paper, the NbN/AlN/NbN device models are created from the atomic level. As we know, it is difficult to fabricate single-crystal AlN thin films in the manufacture, which are primarily polycrystalline or amorphous, and have certain internal defects [25]. Among these defects, the N-vacancy is the most common defect type [25,26]. By studying the microstructure of the all-nitride JJs, the influence of N-vacancy defects on the electrical properties of the junction is explored. The critical factors that affect the conductivity of the models are illustrated from the two aspects, which are the uniformity and position of vacancies distribution in each atom layer. The results show that the uniformity of vacancies has a greater influence on the conductivity of the model. The study, in directly controlling the transport properties of the Josephson junction by N-vacancies, can guide the fabrication of ideal JJs and improve the yield of quantum chips.

## 2. Materials and Methods

### 2.1. Designing Defect Models with N Vacancies in the Barrier

In this study, we constructed a three-layer structure model of NbN/AlN/NbN, with NbN as the electrode and AlN as the central region. Considering that the lattice mismatch between the electrode and the center region should be as low as possible during modeling, three models, as shown in Figure A1, corresponding to different lattice constants, were selected for comparison. Model 3 (in Figure 1a), which has the lowest lattice mismatch degree and the highest binding energy (as shown in Table A1), was selected as the reference model, and its influence on electrical properties was studied by creating N vacancies.

To increase the comparability of the experiments, all defect models had the same total vacancy numbers and the same vacancy element type. The total number of vacancies was 8, and the vacancy element was nitrogen. Based on the reference model 3, all N vacancies were distributed at the interface layer to construct a defect model 3b, as shown in Figure 1b. The defect models 3c–3f, as shown in Figure 1c–f, respectively, were constructed by setting different N vacancies for each atomic layer of AlN in the reference model. Under the condition of ensuring that the number of N vacancies in each atomic layer of AlN in the defect model 3c was consistent, the position of N vacancies in the atomic layer was adjusted, and the models 3c-1com and 3c-com2 were constructed, as shown in Figure 1g,k. The three models were also called the group 3c. There were five groups of models, and we controlled the number of vacancies and changed the positions of vacancies in the same group model, to study the influence of the number and positions of vacancies on the electrical properties, respectively. The N atomic vacancies of model 3b were all on the upper interface, and the vacancy distribution of model 3c–3f was set in the different positions, as shown in Figure 1. The vertically aligned models were the same group comparison models, while the horizontally aligned models were different groups of models in the figure.

### 2.2. Interface Configuration

Miller Bravais indices are an essential parameter affecting interface architecture. The crystal properties of the substrate largely determine the structure of the deposited material. However, there is an interesting phenomenon that AlN shows a (001) orientation and high resistance in the lower layer of NbN sputtering deposition, but the reverse electrode NbN deposited on the AlN barrier layer has a polycrystalline structure [11]. The study indicates that the thickness of the barrier, such as AlN, affects the crystal orientation of the NbN layer at the top of the barrier [19]. Therefore, the thickness of AlN selected in this experiment is within 1 nm to ensure that the upper NbN still has a single crystal structure. The epitaxial growth of NbN on single crystal Si (200) substrate is mainly characterized by (111) and (100) cubic structure, and the AlN film is characterized by (111) hexagonal structure [20,27,28]. There is no primary orientation relationship between NbN and AlN grown under different process conditions. To better compare the orientation in the model, the lattice constant, Miller indices, and other parameters were selected as shown in Table A1.

The interface in the vacancy model was optimized after relaxation, when the model was constructed. The convergence criteria of total energy and force were consistent with the defect-free model (as shown in Section A.2). It was also necessary to carry out quantum transport calculation for the defect models, and the parameter setting of the calculation was consistent with that of the defect-free models (as shown in Section A.2), for which the overall energy of the model tends to be the lowest, and the overall structure of the model tends to be stable.

### 2.3. Model Calculation

After the device models were built, the quantum transport properties were calculated using the non-equilibrium Green’s function (NEGF)-DFT quantum transport program Nanodcal. The density matrix ρ can be given by the intrinsic state $\psi_{i}$ of the KS equation: (1)$$\rho \left(\mu ,\beta \right)\equiv e^{-\beta (H-\mu )}=\sum_{i}|\psi_{i}\rangle f^{EQ}(E-\mu ;\beta )\langle \psi_{i}|$$
where β is the electron temperature, μ is the chemical potential, $f^{EQ}\left(E-\mu ;\beta \right)$ is the Fermi Dirac distribution function, expressed as:(2)$$f^{EQ}\left(E-\mu ;\beta \right)=\frac{1}{1+\exp\beta (E-\mu )}$$
where *E* is the electron energy. To achieve self-consistency, the density matrix is iteratively solved by calculating the Green’s function, which can be expressed as:(3)$$G^{R,A}\left(E\right)={\left[ES-H-\Sigma^{R,A}\right]}^{-1}$$
where $G^{R}$ and $G^{A}$ indicate the retarded and advanced Green’s functions, respectively. *S* is the overlap matrix caused by non-orthogonal orbits (if the orbits are orthogonal, *S* is the identity matrix), and *H* is the Hamiltonian matrix of the system. $\Sigma^{R}$,$\Sigma^{A}$ represents retarded and advanced self-energy respectively, and $G^{A}$ can be calculated by $G^{A}={\left(G^{R}\right)}^{†}$.

Considering an open system with several leads, the current can be calculated by the Landauer formula: (4)$$I=\frac{2e}{h}\sum \int d\varepsilon [f_{l}\left(\varepsilon \right)-f_{r}\left(\varepsilon \right)]T\left(\varepsilon \right)$$
where $f_{l}$, $f_{r}$ is the distribution function of the electrons in the left or right of the electrodes, respectively. The factor of 2 comes from the spin degeneracy.

In addition, the conductance can be defined as
(5)$$G=G_{0}\int d\varepsilon \frac{f_{l}\left(\varepsilon \right)-f_{r}\left(\varepsilon \right)}{eV_{r}-eV_{l}}T\left(\varepsilon \right)$$
where $G_{0}=\left(2e^{2}\right)⁄h$ is the quantum conductance, *h* is Planck’s constant, *e* is the charge of a single electron, and $f_{l}\left(\epsilon \right)-f_{r}\left(\epsilon \right)$ is the difference between the Fermi distribution of the left and right electrodes, $V_{r}-V_{l}$ is the difference in bias voltage applied to the left and right electrodes, and *T(ϵ)* is the transmission coefficient under a certain energy. When the applied bias voltage approaches zero, the limit value obtained is the conductivity value in the equilibrium state. It can be seen from the formula that the conductivity is closely related to the applied bias voltage and the transmission coefficient.

This theory, it should be noted, is based on the conventional ballistic transport theory and calculates the electrical properties of devices at 0 K. The tunneling effect of the Cooper pair at superconducting temperature and the influence of electroacoustic interactions are not considered, so the conductance generated by the tunneling effect of the JJs in the actual process is different. However, from the qualitative point of view, the calculation results have reference value, and in the study of the impact of defects on electrical properties, the first-principles simulation model can be more convenient to guide the process.

## 3. Results and Discussion

### 3.1. Electrical Properties of Defect Models

After optimization, the layer spacing at the interface was changed, as shown in Table 1. At the upper interface, the bond lengths between Al in AlN and N in NbN decreased from 2.09 Å to 1.99 Å. The same happened at the lower interface, where atoms converged near the vacancy [29]. The bond length between the N atom in the barriers and the Nb atom in buffers increased from 2.25 Å to 2.29 Å at the upper interface. Still, there was no obvious pattern at the lower interface. Due to the vacancies, the Al atoms in the interface layer attracted the N atoms in the NbN, and the distance between Al and N became smaller at the upper and lower interfaces. On the contrary, the N atom in AlN repulsed with the Nb atom in NbN, resulting in the bond between N and Nb getting longer, and the lower interface also showed such a rule.

Defects, including the quantity uniformity and position distribution, were observed by increasing N vacancies, and the effects of vacancies were analyzed on the electrical properties of the models. Since the conductance of the three defect-free models is in the same order of magnitude and the difference is slight (as shown in Figure A2), the vacancies were added to study the factors affecting the junction performance. To qualitatively characterize the differences in the conductivity of different vacancy models in the barrier layer, we calculated the conductivity under zero bias voltage, as shown in Figure 2a. It is found that vacancies have a significant influence on the conductance of the models. In Figure 2a, the same color represents the same group of models. It can be seen that vacancies lead to an increase in the conductance of the models. The conductance of model 3f is the largest, while that of model 3b is the smallest. The research also shows that, due to the existence of vacancies, additional electronic states will be formed in the insulator gap [30,31], which may cause an increase in conductance. The data are consistent with the variation of the transmission coefficient shown in Figure 2b, which reflects the transmission coefficient of different vacancy distribution models near the Fermi level in the equilibrium state. It can be seen from the figure that, near the Fermi level, the transmission coefficient of model 3f is the strongest, while that of model 3b is the weakest. Previous studies on oxide vacancies have shown that a single oxygen vacancy or oxygen vacancy chain would lead to delocalization, resulting in a wide transport range and high transmission coefficient [32,33], which was similar to our result.

### 3.2. Electron Density

Local density of states in all of vacancy models near the Fermi level were calculated, as shown in Figure 3, which is consistent with the vacancy distribution shown in Figure 1. In the defect models, the distribution of vacancies in each atomic layer has a great influence on the density of electron states in the barrier layers, as shown in the horizontal view of Figure 3. In each group of models, the number of vacancies in each layer is the same, but the positions are different, resulting in different degrees of change in density of states in the barrier, as shown in Figure 3c,g,k. In general, the density of states in the barrier is the highest in the group 3f, while there is basically no electron state in the barrier layer of the model 3b. It indicates that the possibility of electrons passing through the group 3f is the largest, so the conductivity will be better. This can also explain the transmission coefficient of model 3f being the highest, while that of model 3b is the lowest. On the whole, vacancies lead to an increase in the density of states. Not only the density of states near the Fermi level becomes prominent, but the peak values of the valence band and conduction band increase in varying degrees. It can be seen from Figure 3 that the higher the density of states in the barrier, the higher the corresponding transmission coefficient, which is also similar to the previous research results [29], thus leading to the increase of conductance.

The electronic structure of AlN is closely related to its crystal structure. The results show that the crystal system of zinc blende AlN in metastable phase is hexagonal, which belongs to an indirect band gap semiconductor, while the AlN with zinc blende structure belongs to a direct band gap semiconductor [34]. Model 3b and 3d were selected for the electron density comparison with the defect-free model. The vacancy distribution and conductance of these three models were different. Inspecting the total density of states in the device (as shown in Figure 4a,c,e, respectively), the density of states increases in both defect model 3b and 3d. The local density of states in Figure 3a,b,d can be related to the total density states shown in Figure 4a,c,e, respectively, which reveals that the device models of three structures are metallic. There is no state in the barrier layer at the Fermi surface (as shown in Figure 3a), while the metallicity of the devices in Figure 4a at the Fermi surface should be attributed to the role of NbN, which is consistent with the results of previous studies [23,33]. When the vacancies are made, AlN in the device model generates unsaturated bonds. The conduction band and valence band appear degeneracy and metallicity, increasing the density of states near the Fermi surface (as shown in Figure 4b,c).

It is found that the vacancies lead to the change of the electron density in the vacancy areas. From the diagrams (as shown in Figure 4b), the delta charge of the defect-free model in the bonding region is relatively large, evenly arranged, and has clear boundaries. Due to the absence of N atoms, the bond between Al and N breaks and the dangling bonds appear around Al atoms. The delta charge in the bonding region near the vacancy becomes smaller, and the local deformation occurs (as shown in Figure 3d,f). The delta charge at the N vacancy becomes smaller, indicating that the electrons gather on the Al ion near the N vacancy, causing the change of differential electron density, which is also responsible for enhancing the conductance [32].

Owing to the defects, fluctuations in local density and stoichiometry will lead to local transmission channels. Previous studies have shown that conduction mainly passes through metal channels in highly hypoxic structures [35]. The vacancies of the O atom help the electron to tunnel through the channel of the metal ion [32]. In our study, the absence of N atoms leads to an increase in local density of states near the Fermi level and the formation of local transport channels. The change of delta charge near the N vacancies indicates the aggregation orientation of electrons, which aggregate towards metallic Al atoms to achieve the channel tunneling of electrons through Al ions.

### 3.3. Analyzing Vacancies Distribution

To further analyze the influence of vacancies on the electrical properties of the models, we have made statistics on the vacancy distribution all models. The number distribution of N vacancies is counted in each barrier layer of each model, and the variance of the number distribution is calculated. Compared with the corresponding conductance, the average conductance growth amplitude of the vacancy model with that of the defect-free model is worked out as shown in Table 2. It is clear from the table that the uniformity of vacancy distribution affects the conductance of the model to a certain extent. When the variance of vacancy distribution is minor, that is, the number of vacancies is evenly distributed, and the conductance is relatively large. On the contrary, the conductance is small, as shown in Figure 5a. From the quantity distribution of vacancies, group 3f with the minimum variance has the most considerable conductance, while model 3b with the maximum variance has the smallest conductance. This is due to the uniform distribution of vacancies, which reduces the electron tunneling barrier, increasing the electron tunneling probability and the conductance correspondingly.

It can be seen that the positions of vacancies have little influence on conductance. By comparing models 3c, 3d, 3e, and 3f with their comparative experiments in the same group (as shown in Table 2), it is found that the conductance does not vary significantly, when the number of N vacancies in each layer remains unchanged and the positions are changed. To compare the influence of the position distribution of N vacancies on the conductance, the bond length between each Al atom of the barrier layer and N vacancies in each group of classification models was averaged. Then, the average bond length between Al atoms of the barrier layer and N vacancies in the whole model is obtained (as shown in Figure 5b). In general, the conductance is inversely proportional to the average bond length from the figure. By comparing the mean values of four groups, the mean values of group 3d and 3f become smaller, that is, the more uniform the vacancy distribution is, the average bond length tends to decrease. Meanwhile, the larger the conductance, the larger the average bond length fluctuation.

A large number of previous studies have shown that vacancy defects in oxides have a significant effect on the electrical properties of materials [32,33,35,36,37,38,39]. In our study, vacancy defects enhance the electrical properties of the device model, indicating that nitrides and oxides have similar phenomenal-level defect characteristics. Compared with previous studies, the vacancy distribution was a single vacancy or continuous vacancy chain [32,33], and the distribution was very regular. Our study reveals the effect of the pseudorandom distribution of N vacancies on the electrical properties of materials, indicating that the uniformity of the vacancy distribution in the barrier layer has a significant influence on the electrical properties. At the same time, although AlN is a single crystal material and its structure is relatively simple, its microstructure is still extremely complex in the actual process. From the analysis of the distribution of vacancy positions, we can also see the complexity of its microscopic characteristics. The study of the microstructure of AlN is very worthwhile in reducing the two-level defects of superconducting materials and improving the performance of Josephson junctions.

## 4. Conclusions

The microstructure of the barrier layer in JJs plays an essential role in the performance of the junction. After calculating the conductance of three defect-free models, the model with the highest binding energy and the lowest mismatch rate is taken as the reference model. Considering the effect of vacancies on the junction performance, we characterize the conductance of different defect models at zero bias. Combining the DFT and NEGF, the results show that the vacancies increase the electrical transport properties of the junctions. When the total number of vacancies is controlled and the number of N vacancies is increased in each layer, the conductance is greatly affected. When the number of N vacancies in each layer is fixed and the positions of the vacancies are changed, the conductance is not obviously affected. By calculating the average bond length between each Al atom and N vacancies in the barrier layer and combining with the variance distribution of N vacancies, it is found that the uniformity of vacancy distribution has a great influence on the conductivity of the junction. We mainly guarantee the accuracy of calculation from the perspective of first principles. Although the model has undergone various relaxations and self-consistency of functional calculation, it will still be different from the actual fabrication of JJs. In addition, since the ballistic transport theory is used in the calculation and the electrical properties at 0K are calculated, the tunneling effect of Cooper pairs at the superconducting temperature and the influence of phonons are not considered, and we can only qualitatively discuss the impact of N-vacancy defects on the electrical transport properties of Josephson junctions. However, defects can be inevitable in the process of fabrication. The in-depth understanding of the defect mechanism can help us provide a more precise idea and guidance in the manufactures.

## 5. Patents

The Chinese patent “Method for regulating the transport properties of NbN/AlN/NbN Josephson Junction by using N-vacancy defects” has been accepted.

## Figures and Tables

**Figure 1 nanomaterials-13-00542-f001:**
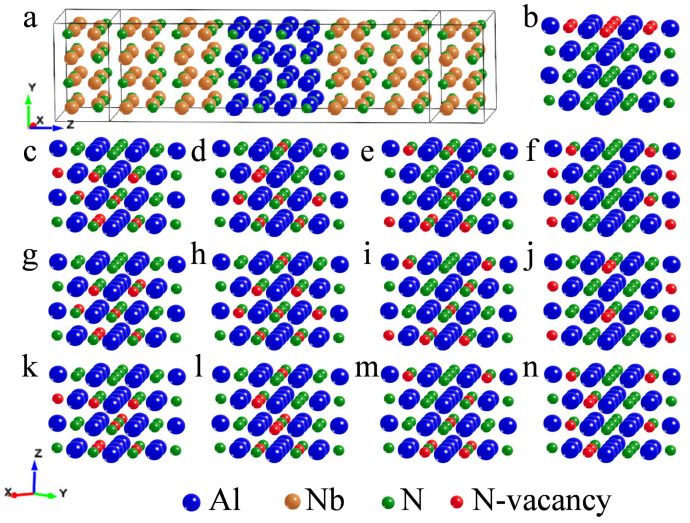
AlN vacancy distribution of barrier layer in model 3. Blue atom is Al, orange atom is Nb, green atom is N, and red atom is N vacancy. (**a**) the atomic structure of the barrier in defect-free model 3; (**b**) the vacancies of model 3b; (**c**,**g**,**k**) the vacancies distribution of the group 3c, including model 3c, 3c-com1, and 3c-com2. The number of atoms in each layer is 0, 3, 2, 3, respectively; (**d**,**h**,**l**) the vacancies distribution of the group 3d, including model 3d, 3d-com1, and 3d-com2. The number of vacancies of atoms in each layer is 1, 2, 3, and 2, respectively; (**e**,**i**,**m**) the vacancies distribution of the group 3e, including model 3e, 3e-com1, and 3e-com2. The number of vacancies of atoms in each layer is 2, 1, 1, and 4, respectively; (**f**,**j**,**n**) the vacancies distribution of the group 3f, including model 3f, 3f-com1, and 3f-com2. The number of vacancies of atoms in each layer is 2.

**Figure 2 nanomaterials-13-00542-f002:**
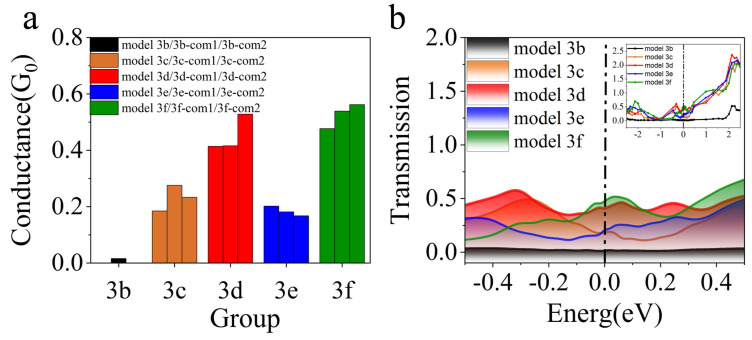
Conductance, transmission spectrum analysis, and local density analysis of different models. (**a**) comparison of zero-bias conductance among different defect models. Bars of the same color represent the same group of models; (**b**) transmission spectrum comparison of different vacancy models near the Fermi level. Inset shows the transmission spectrum from −4 eV to 4 eV.

**Figure 3 nanomaterials-13-00542-f003:**
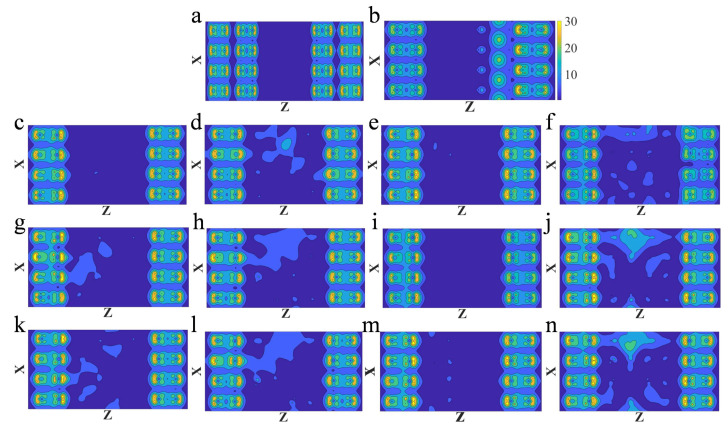
Comparison of local density of states near Fermi level between the defect-free model and different defect models. (**a**) the local density of states in model 3 without any defects; (**b**) the local density of states in model 3b; (**c**,**g**,**k**) the local density of states in the model 3c, 3c-com1, and 3c-com2, respectively; (**d**,**h**,**l**) the local density of states in the model 3d, 3d-com1, and 3d-com2, respectively; (**e**,**i**,**m**) the local density of states in the model 3e, 3e-com1, and 3e-com2, respectively; (**f**,**j**,**n**) the local density of states in the model 3f, 3f-com1, and 3f-com2, respectively.

**Figure 4 nanomaterials-13-00542-f004:**
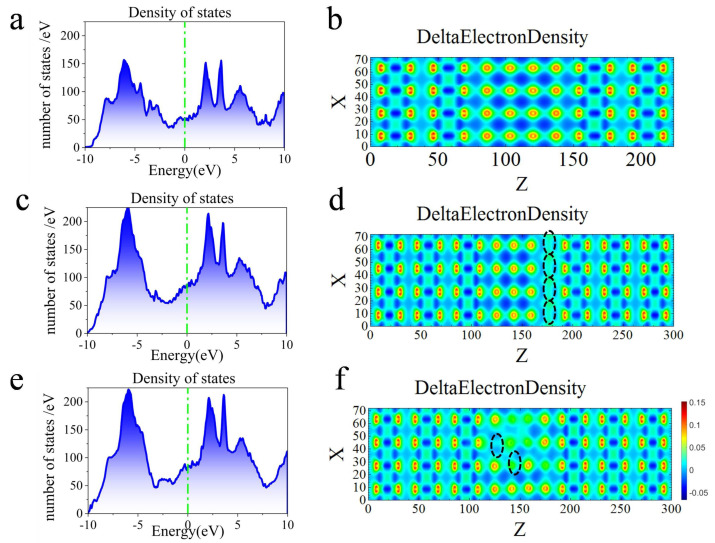
Comparison of the total density of states and delta electron density between typical vacancies model and defect-free model. (**a**,**c**,**e**) total density of states of model 3, 3b, 3d, respectively; (**b**,**d**,**f**) delta electron density of model 3, 3b, and 3d, respectively. The black dotted circle is the position of N vacancies in the surface.

**Figure 5 nanomaterials-13-00542-f005:**
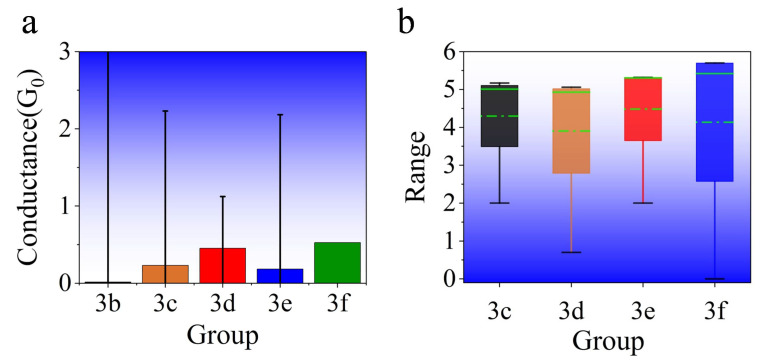
(**a**) Comparison of average conductance among defect models and variance of vacancies in each group. The black line shows the difference of variance; (**b**) box plot of positions distribution in each group. The green dotted line shows the mean value, and the solid line shows the median value.

**Table 1 nanomaterials-13-00542-t001:** Comparison results of the spacing between the layers at the interface before and after relaxation.

Model	The Upper Interface		The Lower Interface	
	Al-N(Å)	N-Nb(Å)	Al-N(Å)	N-Nb(Å)
Relaxation before	2.090	2.248	2.265	2.448
3	1.991	2.292	1.996	2.296
3b	1.982	— *	1.959	2.318
3c	1.928	2.388	1.916	2.406
3c-com1	1.939	2.355	1.916	2.411
3c-com2	1.932	2.413	1.917	2.442
3d	1.930	2.357	1.931	2.349
3d-com1	1.935	2.365	1.939	2.348
3d-com2	1.946	2.382	1.943	2.394
3e	1.914	2.378	1.904	2.398
3e-com1	1.923	2.405	1.912	2.472
3e-com2	1.920	2.389	1.915	2.425
3f	1.938	2.344	1.921	2.446
3f-com1	1.919	2.417	1.927	2.400
3f-com2	1.920	2.408	1.927	2.406

* There is no N atom on the interface of model 3b, and the layer spacing comparison between N atom and Nb atom in the second outer layer is not significant, so this value is not counted here.

**Table 2 nanomaterials-13-00542-t002:** Conductance comparison results of different vacancy models.

Model	Vacancy Distribution	Variance of Vacancy Distribution	Conductance (G0)	Average Growth Rate (%)
3	— *	— *	0.0032	— *
3b	(8, 0, 0, 0)	12	0.0155	3.8
3c	(0, 3, 2, 3)	1.5	0.1845	71.2
3c-com1	(0, 3, 2, 3)	1.5	0.276	
3c-com2	(0, 3, 2, 3)	1.5	0.2328	
3e	(2, 1, 1, 4)	1.5	0.2018	56.3
3e-com1	(2, 1, 1, 4)	1.5	0.1813	
3e-com2	(2, 1, 1, 4)	1.5	0.1671	
3d	(1, 2, 3, 2)	0.5	0.4138	140.4
3d-com1	(1, 2, 3, 2)	0.5	0.4157	
3d-com2	(1, 2, 3, 2)	0.5	0.5278	
3f	(2, 2, 2, 2)	0	0.4767	163.3
3f-com1	(2, 2, 2, 2)	0	0.5386	
3f-com2	(2, 2, 2, 2)	0	0.5621	

* Model 3 is the reference defect-free model with no vacancies.

## Data Availability

The data that support the findings of this study are available from the corresponding author upon reasonable request.

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
