# Peer review of "Directly Controlling the Transport Properties of All-Nitride Josephson Junctions by N-Vacancy Defects"

_nanomaterials, 2023, doi:10.3390/nano13030542_

Round 1
Reviewer 1 Report
The manuscript presents a concrete discussion of the influence of the N-vacancy on electrical conductance, in the aspects of transmission coefficient, density of electron states, crystal structure, and even the vacancy location. With clear statements of the results, I suppose this manuscript is ready to publish.
A minor revision, please give a space of the words between ‘Model’ and ‘3b’ in Page 7 line 177.
And I would like to ask two questions:
1. Page 6 Line 146 - 147, are the additional electronic states mentioned here corresponding to the increase of the density of states in the later discussion in section 3.2 Electron density? Can one claim that all the additional electron states here contribute to the enhancement of the electrical conductance?
2. The defects discussed in the manuscript are based on the vacancy defects, as given in the title, can the authors also provide some prediction about the conductance influence from the antisite defect?
Author Response
We would like to thank you for your careful reading, helpful comments, and constructive suggestions, which has significantly improved the presentation of our manuscript.
We have carefully considered all comments from the reviewers and revised our manuscript accordingly. In the following section, we summarize our responses to each comment from the reviewers. We believe that our responses have well addressed all concerns from the reviewers. We hope our revised manuscript can be accepted for publication.
Comment 1:
A minor revision, please give a space of the words between ‘Model’ and ‘3b’ in Page 7 line 177.
Our reply:
We have modified it, thank you for your care.
Corresponding change in manuscript: Yes
Location of Change: Page 7 line 177
Comment 2:
Page 6 Line 146 - 147, are the additional electronic states mentioned here corresponding to the increase of the density of states in the later discussion in section 3.2 Electron density? Can one claim that all the additional electron states here contribute to the enhancement of the electrical conductance?
Our reply:
Yes, the additional electronic states mentioned here are corresponding to the increase of the density of states in the later discussion in section 3.2 Electron density.
We can say: the additional electron states contribute to the enhancement of the electrical conductance, but not all. And we also think that the additional electronic states may have a contribution from the superconducting layer (NbN layer). Compared with Fig. 4a, the electron states in Fig. 4c and Fig. 4e do increase at the Fermi level due to the vacancy, that is to say, the vacancy adds additional electron states. We conclude that some of the additional electron states contribute to the enhancement of the electrical conductance.
Comment 3:
The defects discussed in the manuscript are based on the vacancy defects, as given in the title, can the authors also provide some prediction about the conductance influence from the antisite defect?
Our reply:
We are sorry that our models can not predict the antisite defect at present. Thank you very much for your question. We will consider this problem in our subsequent experiments and further explore this problem.
Thank you again for your positive review of this paper, and at the same time, your questions have triggered us to think more deeply, thank you very much!

Reviewer 2 Report
In this work, authors have applied the first principles and non-equilibrium Green’s function to calculate the electrical transport characteristics of the preset model. And shows the importance of the uniformity of vacancy distribution in resulting in a uniform number of vacancies can effectively increase the conductance of Josephson junction. This work clarifies the effect of the N-vacancy defects on the conductivity of the NbN-based Josephson junctions, which offers helpful guidance for understanding the microscope mechanism of the NbN-based Josephson junction. This article will be important in improving the yield of superconducting quantum chips in the future. Therefore, I recommend the paper for publication in nanomaterials with miner comments below.
Comment 1.
It looks typo mistake on page 2, lines 67 and 69. The figure number and table number are also missing on pages 4 and 5, lines 101 and 139.
Comment 2.
The interface thickness is an essential parameter for the Josephson junction. How much was the interface thickness of the AlN-NbN-Substrate taken in this model?
Comment 3.
The number of defects can improve the Junction's performance; if so, any limit on the number of defects.
Comment 4.
Does this model give any idea? What will be effect by the superconducting properties of the NbN layer with the increasing number of defects?
Author Response
We would like to thank you for your careful reading, helpful comments, and constructive suggestions, which has significantly improved the presentation of our manuscript.
We have carefully considered all comments from the reviewers and revised our manuscript accordingly. In the following section, we summarize our responses to each comment from the reviewers. We believe that our responses have well addressed all concerns from the reviewers. We hope our revised manuscript can be accepted for publication.
Comment 1:
It looks typo mistake on page 2, lines 67 and 69. The figure number and table number are also missing on pages 4 and 5, lines 101 and 139.
Our reply:
We have modified it in the paper. This problem should be due to problems when latex links to pictures and tables after deleting the content of the supplementary document. Meanwhile, we ignored this problem when converting pdf to word. Thank you for your care.
Corresponding change in manuscript: Yes
Location of Change: Page 2 line 67,69
Page 4 line 101
Page 5 line 139
Comment 2:
The interface thickness is an essential parameter for the Josephson junction. How much was the interface thickness of the AlN-NbN-Substrate taken in this model?
Our reply:
The thickness of the barrier layer has been mentioned in the paper. On page 4, lines 95 and 96. The exact thickness is 0.6nm.
Comment 3:
The number of defects can improve the Junction's performance; if so, any limit on the number of defects.
Our reply:
Thank you very much for asking this question, and that's the goal of our next experiment. In this round of experiments we have a different number of vacancies in each layer. In the present experiment, the limiting factor of vacancy is not clear enough. In the next step, we will set the different total number of vacancies in different positions, and continue to explore the influence of the number of vacancies on conductance. After the experiment is completed, we will be able to get the limiting factors of vacancies.
Comment 4:
Does this model give any idea? What will be effect by the superconducting properties of the NbN layer with the increasing number of defects?
Our reply:
This paper focuses on the effect of the number distribution and position distribution of N-vacancy in each atomic layer on the electrical properties of the junction. How the vacancy affects the process is the focus of our research group in the next step. We hope to accurately control the electrical properties of the junction while studying the defects. Of course, this is not possible with the current model, but it is our long-term goal.
At present, because the material in the electrode region (NbN layer) in the theory has little influence on the performance of the whole junction, that is to say, the conclusion of this paper is still applicable theoretically with other superconducting materials. We have not yet done relevant research on the influence of increasing vacancy on the superconducting layer. In this aspect, we can further conduct additional experiments to explore. Thank you for your question.
Thank you again for your positive review of this paper, and at the same time, your questions have triggered us to think more deeply, thank you very much!
